# Analogs of 6-Bromohypaphorine with Increased Agonist Potency for α7 Nicotinic Receptor as Anti-Inflammatory Analgesic Agents

**DOI:** 10.3390/md21060368

**Published:** 2023-06-20

**Authors:** Igor A. Ivanov, Andrei E. Siniavin, Victor A. Palikov, Dmitry A. Senko, Irina V. Shelukhina, Lyubov A. Epifanova, Lucy O. Ojomoko, Svetlana Y. Belukhina, Nikita A. Prokopev, Mariia A. Landau, Yulia A. Palikova, Vitaly A. Kazakov, Natalia A. Borozdina, Arina V. Bervinova, Igor A. Dyachenko, Igor E. Kasheverov, Victor I. Tsetlin, Denis S. Kudryavtsev

**Affiliations:** 1Shemyakin-Ovchinnikov Institute of Bioorganic Chemistry, Russian Academy of Sciences, 117997 Moscow, Russiavictortsetlin3f@gmail.com (V.I.T.); 2N.F. Gamaleya National Research Center for Epidemiology and Microbiology, Ivanovsky Institute of Virology, Ministry of Health of the Russian Federation, 123098 Moscow, Russia; 3The Branch of the Shemyakin-Ovchinnikov Institute of Bioorganic Chemistry, Russian Academy of Sciences, 142290 Pushchino, Russia; 4Center Neurobiology and Brain Restoration, Skolkovo Institute of Science and Technology, 121205 Moscow, Russia; 5Center of Life Sciences, Skolkovo Institute of Science and Technology, 121205 Moscow, Russia; 6Department of Biology, M.V. Lomonosov Moscow State University, 119991 Moscow, Russia; 7Moscow Institute of Physics and Technology, 141701 Dolgoprudny, Russia

**Keywords:** α7 nicotinic acetylcholine receptor, nAChR, agonist, ligand, inflammation, potency, hypaphorines

## Abstract

Hypaphorines, tryptophan derivatives, have anti-inflammatory activity, but their mechanism of action was largely unknown. Marine alkaloid L-6-bromohypaphorine with EC_50_ of 80 μM acts as an agonist of α7 nicotinic acetylcholine receptor (nAChR) involved in anti-inflammatory regulation. We designed the 6-substituted hypaphorine analogs with increased potency using virtual screening of their binding to the α7 nAChR molecular model. Fourteen designed analogs were synthesized and tested in vitro by calcium fluorescence assay on the α7 nAChR expressed in neuro 2a cells, methoxy ester of D-6-iodohypaphorine (6ID) showing the highest potency (EC_50_ 610 nM), being almost inactive toward α9α10 nAChR. The macrophages cytometry revealed an anti-inflammatory activity, decreasing the expression of TLR4 and increasing CD86, similarly to the action of PNU282987, a selective α7 nAChR agonist. 6ID administration in doses 0.1 and 0.5 mg/kg decreased carrageenan-induced allodynia and hyperalgesia in rodents, in accord with its anti-inflammatory action. Methoxy ester of D-6-nitrohypaphorine demonstrated anti-oedemic and analgesic effects in arthritis rat model at i.p. doses 0.05–0.26 mg/kg. Tested compounds showed excellent tolerability with no acute in vivo toxicity in dosages up to 100 mg/kg i.p. Thus, combining molecular modelling and natural product-inspired drug design improved the desired activity of the chosen nAChR ligand.

## 1. Introduction

Homopentameric α7 nicotinic acetylcholine receptors (nAChRs) are ligand-gated cationic ion channels with high permeability for calcium ions and a fast desensitization (see reviews [1,2]). They are widespread in the nervous system, mainly in areas responsible for cognitive function and memory—such as the hippocampus, cerebral cortex, and several subcortical structures of the limbic system [3]. Dysfunction of α7 nAChR is associated with neuropsychiatric and neurological disorders such as schizophrenia and Alzheimer’s disease [4,5]. α7 nAChRs are also expressed in non-neuronal cells, in particular, in immune and endothelial cells where they play a decisive role in at least some types of inflammation [6]. These receptors are also involved in neuroprotection in astrocytes, oligodendrocyte precursor cells, and microglia [7,8,9].

In non-neuronal cells, including lymphocytes, dendritic cells, and macrophages, α7 nAChR is an essential participant in the cholinergic anti-inflammatory pathway, which is the link between the innate immune system and efferent nerves [10]. As well, α7 nAChR mediates the cholinergic pathway in brain, which is crucial in neuroprotection and, probably, in Parkinson’s disease, oxygen deprivation, and global ischemia [9,11].

Intracellular signaling, mediated by the activation of α7 nAChR, is based on conducting short-term currents of Na^+^, K^+^, and Ca^2+^ ions, the mechanism of calcium-induced calcium release through the IP3 receptors being one of the essential ones. In certain cases, α7 nAChRs function as metabotropic receptors, mediating intracellular signals by binding to G proteins (Gα and Gβγ) [12,13]. In immune cells, α7 nAChR is involved in several intracellular pathways resulting in the anti-inflammatory effects [14]. For example, α7 nAChR has been shown to activate JAK2/STAT3 cascade, which leads to inhibition of the transcription factor NF-κB and to the production of anti-inflammatory cytokines [15,16]. It has also been shown that α7 nAChR activates the PI3K/Akt pathway, which promotes Nrf-2 translocation into the nucleus and an increase in heme oxygenase (HO-1) expression, which leads to potent anti-inflammatory effects [11,17,18,19].

Thus, the activation of α7 nAChR is considered a potential therapeutic strategy against neurological and inflammatory disorders [14,20]. The search for new agonists, especially selective ones, is especially perspective. The α7 nAChR agonist EVP-6124 has shown positive results in clinical trials for the treatment of conditions such as schizophrenia [21], but trials were discontinued due to side effects from the digestive system; ABT-594 is a less toxic analog of nicotine and has analgesic properties [22]. Anabasein analog GTS-21 is another example of analgesic and anti-inflammatory activity of α7 nAChR agonists (see Section 2.4 in review [23]). Unfortunately, no α7 nAChR agonists have passed phase III of clinical trials as analgesic, anti-inflammatory, or anti-psychotic means.

Several years ago, we found that 6-bromohypaphorine (6BHP) from the nudibranch Hermissenda carassicornis acts as an agonist of human α7 nAChRs [24]. Currently, anti-inflammatory activity of its parent compound hypaphorine is described in detail [25,26,27,28]. Interestingly, acetylcholine esterase-blocking activity has been demonstrated recently for L- and D-hypaphorines [29]. Moreover, intracranial hypaphorine induces sleep in mice [30], and cholinergic system is known to be involved in the sleep–wake cycle regulation [31]. Thus, we decided to check the anti-inflammatory and analgesic activities of 6BHP and to try to increase its affinity for the α7 nAChRs by designing its new analogs.

In the present communication, we explored diverse 6BHP synthetic analogs described by the general formula depicted on Figure 1, where X = NO_2_, NH_2_, OH, OCH_3_, Cl, Br, I; Q = OH, OCH_3_, OC_2_H_5_, NH_2_, NHCH_3_; R = H, CH_3_.

## 2. Results

### 2.1. Rational Design and Virtual Screening

The potency of naturally occurring 6-bromohypaphorine (~90 μM) is considerably lower than that of the traditional α7 nAChR agonists acetylcholine (8 μM) and epibatidine (15 μM). However, it is selective for α7 nAChR [24], which makes it a promising candidate for the design of structurally similar ligands with a higher affinity for α7 nAChRs. The original compound has a relatively simple structure, a small number of rotatable bonds and, in general, similar features with other agonists of nicotinic receptors: quaternary amine and aromatic moiety. Thus, it was possible to develop trustworthy models of complexes of 6-bromohypaphorine and analogs thereof with α7 nAChR using OpenBabel script to generate SMILES [32,33]. Indeed, docking of 6-bromohypaphorine to AChBP/α7 nAChR chimera (PDB 3SQ6) and the recently published cryo-EM structure of α7 nAChR in complex with epibatidine (PDB 7KOX) show binding at the orthosteric site under the loop C (Figure 1a–c). Moreover, 6BHP analogs and epibatidine appear to occupy similar positions in the orthosteric site showing remarkable similarity in the binding modes (Figure 1d,e).

It was previously found that hydrophilic amino acid residues in orthosteric binding site influence agonist binding by interaction with aromatic moiety of the later [35]. Interestingly, carboxylic group of hypaphorine analogs might fit in this interaction and add some subtype selectivity.

### 2.2. Synthesis of a Hypaphorine Analog Series 

Modification of indole core was performed by Sandmeyer reaction in Filimonov modification [36]. See Figure 2 for the illustration. The key intermediate protected 6-aminoindoline derivate was easily synthesized by tryptophan’s selective nitration [37], followed by total protection and reduction. After introducing an appropriate substitution at the 6th position, the indolyl ring was quantitatively reoxidized by 2,3-dichloro-5,6-dicyanobenzoquinone (DDQ). The obtained protected tryptophan derivate was easily converted to the corresponding hypaphorine derivate.

6-Nitro-L-tryptophan. To a stirred solution of 10.2 g (50 mmol) L-tryptophan in 100 mL of glacial acetic acid, 1.9 mL of nitric acid was added. The reaction mixture was cooled to 5 °C, followed by dropwise addition of 4.4 mL nitric acid solution in the 20 mL of glacial acetic acid. After 48 h stirring at room temperature (RT), the precipitate was filtered and washed by cold acetic acid, followed by thorough washings with acetone and drying. The resultant solid was dissolved in 15 mL of saturated sodium acetate solution, and the precipitate was washed and dried under P_2_O_5_. As a result, 2.6 g (21%) of 6-nitro-L-tryptophan was obtained as a yellow fluffy powder. 1H NMR (500 MHz, Deuterium Oxide, pH 12) δ 7.69 (d, J = 7.9 Hz, 1H), 7.60 (d, J = 8.0 Hz, 1H), 7.27 (s, 1H), 7.05 (t, J = 8.0 Hz, 1H), 3.16 (t, J = 7.0 Hz, 1H), 3.03 dd, J = 14.3, 6.6 Hz, 1H), 2.86 (dd, J = 14.3, 7.4 Hz, 1H).

6-Nitro-L-tryptophan methyl ester hydrochloride. To a cooled to −15 °C methanol, 2.5 mL of thionyl chloride was slowly added, and after that, a 1 g of 6-nitro-L-tryptophan was added, mixed to a complete dissolution of solids and warmed to RT. After that, the mixture was heated to the reflux for 1 h and subsequently evaporated under vacuum. The residue was suspended in diethyl ether and filtered. The solid was dried under vacuum to receive 1.1 g (yield 91%) 6-nitro-L-tryptophan methyl ester hydrochloride.

Nα,Nin-di-Boc-6-nitro-L-tryptophan methyl ester. To a suspension of 8.0 g 6-nitro-L-tryptophan methyl ester hydrochloride in 200 mL acetonitrile, 6.2 g of Boc anhydride was added under stirring, followed by portion-wise N,N-Diisopropylethylamine (DIPEA) (5 mL) addition. After 15 min stirring, the second equivalent of Boc anhydride, followed by 350 mg 4-Dimethylaminopyridine (DMAP) was added, and the reaction mixture was stirred overnight and then evaporated. The residue was dissolved in ethyl acetate, washed triple with the citric acid solution, then twice with brine, dried over anhydrous sodium sulphate and evaporated to afford 11 g (89%) of Nα,Nin-di-Boc-6-nitro-L-tryptophan methyl ester. 1H NMR (700 MHz, Chloroform-d) δ 9.11 (s, 1H), 8.19 (dd, J = 8.7, 2.1 Hz, 1H), 7.72 (s, 1H), 7.65 (d, J = 8.7 Hz, 1H), 5.22 (d, J = 7.8 Hz, 1H), 4.71 (d, J = 7.8 Hz, 1H), 3.78 (s, 3H), 3.42–3.34 (m, 1H), 3.24 (d, J = 13.5 Hz, 1H), 1.77 (s, 9H), 1.56–1.42 (m, 9H).

The solution of 6.0 g Nα,Nin-di-Boc-6-nitro-L-tryptophan methyl ester in 150 mL of methanol was hydrogenated for 12 h over 800 mg of 10% Pd/C under the pressure of hydrogen 1 MPa, after that the catalyst was filtered with celite, and the filtrate was evaporated under vacuum to afford the 5.9 g of Nα,Nin-di-Boc-6-nitro-L-dihydrotryptophan methyl ester as a cream foam. 1H NMR (700 MHz, Chloroform-d) δ 7.13 (dd, J = 7.9, 1.7 Hz, 0H), 7.02 (d, J = 7.9 Hz, 0H), 5.16 (s, 1H), 4.25–4.14 (m, 1H), 3.82 (d, J = 2.3 Hz, 3H), 3.78 (s, 1H), 3.73 (ddd, J = 14.4, 9.7, 5.6 Hz, 1H), 3.41 (tt, J = 9.8, 5.4 Hz, 1H), 2.28 (dt, J = 13.1, 5.6 Hz, 1H), 2.12–2.00 (m, 1H), 1.91 (dt, J = 14.0, 8.4 Hz, 1H), 1.73 (s, 2H), 1.64 (d, J = 5.1 Hz, 9H), 1.53 (d, J = 4.4 Hz, 9H).

### 2.3. Functional Assay on α7 nAChR

A panel of the synthesized compounds was tested to determine their activity on α7 nAChR by fluorescence detection of cytoplasmic calcium rise (Figure 2a). All modified hypaphorines (i.e., N,N,N-trimethyl tryptophan derivatives) have shown agonistic activity in the presence of the positive allosteric modulator PNU 120,596 in the range from sub-micromolar to high micromolar concentrations. L-hypaphorine without additional modifications in the indole ring failed to activate α7 nAChR in these conditions at concentrations up to 1 mM, whereas methyl ester of L-hypaphorine showed agonistic activity with EC_50_ between 18 and 37 μM. 6-bromohypaphorine as previously reported shows agonistic activity in high micromolar concentrations [24]. 6-nitrohypaphorine does not show any agonistic activity at concentrations up to 1 mM, however methyl ester of 6-nitrohypaphorine activates α7 nAChR with EC_50_ of 14 μM. These observations suggest that the presence of indole ring substitutions as well as esterification of the carboxy group are crucial to develop α7 nAChR agonist on the basis of hypaphorine structure (Figure 2b). None of N,N-dimethyltryptophan derivatives have shown α7 nAChR agonistic activity (Table 1).

To explore the space of possible substitutions in the hypaphorine molecule (Figure 2b), a systematic approach has been used. Table 1 summarizes the findings of in vitro screening based on fluorescent calcium detection. The most striking effect on the activity showed a variation of carboxylic group modification. Substitutions in the indole ring (denoted as “X” on Figure 2b) consisted of halogens, hydrogen bond acceptors, or hydrogen bond donors. Halogens showed the most positive impact on the tested compound potency, the iodine-substituted one being the most potent (EC_50_ 611 nM). Contrary to that, hydrogen bond donor groups (NH_2_ and OH), when placed in the sixth position of the hypaphorine indole ring, resulted in very low potency with EC_50_ from 52 to 599 μM, respectively. Hydrogen bond acceptors in the indole ring sixth position showed intermediate results. Thus, using rational design methods to get new ligands for α7 nAChR, it was possible to significantly improve their potency for this receptor.

### 2.4. Inhibition of α3-Containing (α3*) nAChRs by Hypaphorine Derivatives

Two of the studied compounds with high and medium agonist potency for α7 nAChR (6CF and 6ID, Table 1) were tested for activity toward α3* nAChR by calcium imaging. Neither 6ID, which was most active agonist of α7 nAChR, nor 6CF showed detectable agonism on α3* nAChR at concentrations up to 30 µM. Test results showed their antagonistic effect on human α3-containing nAChRs (Figure 3). The most effective was (D)-6-trifluoromethylhypaphorine methyl ester (6CF, IC_50_ = 2.4 ± 1.2 µM), least active was (D)-6-iodohypaphorine methyl ester (6ID, IC_50_ = 4.4 ± 1.0 µM).

It should be noted that fluorescent calcium assay results cannot exclude the possibility of partial agonism of the tested compounds. One can imagine such weak activation of the α3* nAChR that does not produce calcium response strong enough to be detected by the plate reader, but will desensitize the receptor (thus, diminishing cellular response to nicotine). Moreover, partial agonists can occupy orthosteric sites and prevent nicotine binding, which would appear as antagonism under these conditions. Further investigation using electrophysiology might resolve this question.

### 2.5. Effects of PNU 282987, Hypaphorine Methyl Ester, and D-6-Iodohypaphorine Methyl Ester (6ID) on the Expression of Macrophage Markers

Basing on our recent work [14], where for the first time activation of α7 nAChR on macrophages by selective agonist PNU 282,987 was shown to affect the expression of various membrane proteins, here, the action of the α7 nAChR agonists was compared with that of parental hypaphorine (not interacting with nAChRs) and with 6ID (our analog with the highest affinity for α7 nAChR) on the expression of macrophage membrane markers CD11b and CD11c. The selective agonist PNU 282,987 increased CD11b and CD11c protein expression by an average of 10% and 26%, respectively (Figure 4a,b). There was some tendency towards an increase in CD11c membrane protein expression under the action of 6ID, however, not statistically significant at *p* < 0.05 (one-way ANOVA, Tukey post hoc test).

The effect of the above-listed compounds on the expression of the TLR4 receptor and the CD86 protein was studied by flow cytometry after 48 h incubation with macrophages. The interaction of bacterial endotoxins with the TLR4 receptor is known to activate the expression of several pro-inflammatory cytokine genes [38], while the co-stimulatory molecule CD86 plays an important role in suppressing inflammatory responses [39]. A decrease in the expression of the TLR4 receptor was observed under the action of all tested compounds (Figure 4c,e), the rank order of these TLR4-suppressing effects was as follows: hypaphorine (14%) < PNU 282,987 (21%) < 6ID (31%).

In addition, we observed an increase in the expression of CD86 under the action of hypaphorine methyl ester (26%) and 6ID (21%), but not PNU 282,987 (Figure 4d,f).

### 2.6. Involvement of ERK and STAT3 in the Protective Role of PNU 282987, Hypaphorine, and 6ID in LPS-Mediated Inflammation in Macrophages

To gain an insight into the signaling induced by the tested compounds, ERK1/2 and STAT3 phosphorylation was checked. Macrophages were treated with tested compounds for 1 h, after which they were stimulated with LPS for 3 h.

Using flow cytometry, it was determined that application of all compounds resulted in a marked increase in the ERK 1/2 phosphorylation (Figure 5a). ERK 1/2 phosphorylation peaked upon 6ID stimulation (MFI cntr vs. 6ID: 194 vs. 274; *p* ˂ 0.001, see Figure 5c). Activation of the STAT3 pathway was less pronounced. However, stimulation with all tested compounds led to a statistically significant increase in the phosphorylation of STAT3 (Figure 5b). 6ID led to the maximum level of STAT3 phosphorylation in macrophages (MFI cntr vs. 6ID: 49 vs. 58; *p* < 0.01).

### 2.7. CFA-Induced Inflammation Test

Anti-inflammatory effect of two analogs of 6-substituted hypaphorine (6ND and 6ID) was tested in a model with subplantar administration of complete Freund’s adjuvant (CFA) in mice. CFA was injected once into the footpad to induce oedema in 24 h. Oedema was measured with an electronic calliper. Immediately after paw measurement, 6ND or 6ID was injected intramuscularly in doses 1, 0.5, and 0.1 mg/kg. Paws were measured at 6, 12, 24, and 48 h after administration of 6ND or 6ID. Only the dose 0.5 mg/kg of 6ND significantly reduced oedema (Figure 6a), whereas 6ID was effective against oedema at doses 0.5 and 1 mg/kg (Figure 6b). Hypaphorine, which did not show α7 nAChR agonistic activity in concentrations up to 1 mM in fluorescent Ca2+ detection (Table 1), has nonetheless demonstrated anti-oedemic properties (Figure 6c). End points at 48 h of drug treatments at dose 0.5 mg/mL have been analyzed in terms of paw diameter normalized to the maximal oedemic levels in each group (Figure 6d). Hypaphorine showed markedly less effective reduction of oedema than 6ND and 6ID.

Additionally, after 2 h following the drug injection, the analgesia test was conducted on the “hot plate” preheated to 53 °C. The time of withdrawal or licking of the affected paw was assessed for each animal. According to the hot plate test results, it was found that 6ND at a dose of 0.1 mg/kg has a pronounced analgesic effect, but higher doses did not show a difference from the control group (Figure 6c). 6ID at doses 0.5 and 1 mg/kg showed significant analgesia (Figure 6d).

### 2.8. Carrageenan-Induced Inflammation Test

Subplantar 1% carrageenan injections in a volume of 30 μL into the right hind paw were carried out to generate an inflammation model in mice. The paw volume was measured beforehand using an electronic calliper. The maximum effect of the introduction of carrageenan was observed after 3 h. One hour after carrageenan injection, the drugs (hypaphorine, 6ND, or 6ID) were administered at a dose of 0.5 mg/kg. Hypaphorine and 6ID were also administered at 1 mg/kg. The control group was treated with saline (0.9% NaCl). Two hours after the drug administration (total 3 h from carrageenan injection), the size of the pad of the right hind paw was measured, the von Frey test and paw pressure analgesimeter pain sensitivity assessment were performed.

Carrageenan injections provoked marked oedema in the absence of any treatment (Figure 7a–c, “control” box plots). In all treatment groups, oedema was reduced, although not to a baseline (Figure 7a–c). However, data normalization to the average paw diameter in each treatment group revealed that effects of 6ND (Figure 7d) and 6ID (Figure 7e) were statistically significant, whereas hypaphorine showed only a tendency to decrease oedema, not statistically significant (Figure 7f, *p* = 0.131, one-way ANOVA, *n* = 8).

All three compounds showed statistically significant analgesia in paw pressure tests (Figure 7g), but no statistically significant difference between tested compounds were detected. Similarly, von Frey test revealed significant analgesia in the case of 6ND, 6ID, and hypaphorine, compared to vehicle, but differences between the compounds were not significant (Figure 7h). Thus, anti-oedemic effect seems to depend on α7 nAChR activity to a greater extent than the analgesic activity of the compounds.

### 2.9. Monosodium Iodoacetate-Induced Arthritis Model

Osteoarthritis is a condition characterized by articular cartilage degradation and activation of inflammation molecular signaling pathways [40]. One of the well-established osteoarthritis animal models is monosodium iodoacetate (MIA) intra-articular injection [41], which was used in the present study to evaluate the perspectives of hypaphorine analogs in anti-inflammatory osteoarthritis treatment.

We previously have published the results of 6ID and 6ND tests in myocardial infarction model, showing significant positive effect of 6ND [42]. In the current communication, we chose 6ND for the test on osteoarthritis rat model.

By the 8th day of the experiment, the maximum inflammation of the knee joint was noted. The intramuscular administration of compound 6ND in the studied doses was carried out from the 8th day of the study to the 16th day. On the 8th and 16th days, the effect of the compound was assessed using a set of functional tests: weight distribution between the healthy and inflamed limbs immediately after the first 6ND injection (Figure 8a) and after eight days of 6ND administration (Figure 8b), muscle strength grasping the hind limbs after the first 6ND injection (Figure 8c) and after eight days of 6ND administration (Figure 8d), mechanical allodynia after the start of 6ND administration and after eight days of 6ND administration (Figure 8e,f, respectively), thermal hypersensitivity 2 h after administration of 6ND (Figure 7g), and measuring the diameter of the knee joint (Figure 8h).

According to the tests, 6ND did not influence body weight distribution between healthy and affected limbs, but significantly improved grip strength of affected limb after eight days of *i.p.* administration (Figure 8d). 6ND exhibited analgesic (anti-allodynic) effect (Figure 8f) and anti-oedemic effect on the MIA-injected joint (Figure 8g), but did not influence thermal hyperalgesia in hot plate test in this model.

### 2.10. Histological Study

After the experiment termination and euthanasia of the experimental animals, biomaterials were taken for histological examination. Soft tissues around the right knee joint were excised as much as possible. Femur, tibia, and fibula were cut across in the middle of the diaphysis closer to the corresponding articular surfaces. The joint was fixed in a 10% solution of neutral formalin for seven days, washed in running water and then decalcified in Trilon B for 7–14 days. After satisfactory decalcification of the bone and cartilage tissue, the joint was cut in the sagittal plane. The diaphysis of the tubular bones was shortened to the border of 2–3 mm from the metaepiphyseal cartilage. In this form, the biomaterial was repeatedly washed in running water, dehydrated in alcohols of ascending concentration, and embedded in paraffin. Paraffin sections 5–7 μm thick were stained with hematoxylin and eosin and examined using conventional light microscopy. During histological analysis, the following morphological signs were assessed: inflammatory infiltration of the synovial membrane (synovitis), synovial hyperplasia, destructive changes in the articular cartilage, and destructive changes in bone tissue (if any). The following scale was used to assess the severity of a particular morphological trait: 0 points—within normal limits, 1—minimal severity, 2—weak, 3—medium (moderate), 4—strong, 5—very strong [43].

After intra-articular administration of MIA, all animals on the 15th day showed characteristic signs of arthritis—inflammatory infiltration of the synovial membrane (synovitis) with signs of its hyperplasia, destructive changes in the articular cartilage of the femur and tibia, as well as destructive and necrotic changes in the menisci. Thus, to the described picture on the 15th day of the experiment, the term “MIA-induced arthritis” can be fully used.

In the group of animals that received saline after intra-articular administration of MIA, a pronounced inflammatory infiltration of the synovial membrane was observed on the 15th day (mean score 3.67), accompanied by synovial hyperplasia (2.33). Destructive changes in the epiphyseal articular cartilage of the femur (3.0) and tibia (3.33) were observed (Figure 9a,b).

The administration of 6ND at a dose of 0.05 mg/kg did not change the inflammatory infiltration of the synovial membrane of the right knee joint of female rats (Figure 9c,d) compared to the control group on the 15th day of observation (average score 3.75, versus 3.67 in control). The mean score for synovial hyperplasia among the animals of this group was estimated as 2.25. Destructive changes in the epiphyseal articular cartilage of the femur and tibia (3.0 and 3.5, respectively) were comparable to those in the group of animals with MIA-induced arthritis treated with saline in a volume of 2 mL/kg.

The administration of 6ND in the model of MIA-induced arthritis at a dose of 0.26 mg/kg contributed to a significant reduction in the morphological manifestations of synovitis and synovial hyperplasia of the right knee joint of female rats: the average score of inflammatory infiltration of the synovial membrane corresponded to 2.5, synovial hyperplasia–1.5 (see Figure 9e,f).

The minimum, average, and maximum estimated scores of synovitis, synovial hyperplasia, and destructive changes in the epiphyseal articular cartilage of the femur and tibia for each experimental group of animals are presented in Table 2 and Table 3.

## 3. Discussion

6-Bromohypaphorine (6BHP) isolated from the marine mollusk *Hermissenda* sp. acts as a silent or partial agonist of α7 nAChR [24]. It was previously shown that activation of α7 nAChR by its selective agonist PNU 282,987 leads to an increase in the expression of macrophage membrane proteins, including HLA-DR, CD11b, and CD54, but reduces the expression of the CD14 receptor [14]. These macrophage markers play a major role in the complex pathogenesis of sepsis, inflammation, and immunosuppression. In current study, using computer-aided design, molecular modelling, and synthesis of fourteen 6-bromohypaphorine analogs, we approached the new class of α7 nAChR agonists and explored their anti-inflammatory and analgesic properties. Unmodified hypaphorine did not show agonistic properties at α7 nAChR (Table 1) but was described previously as anti-inflammatory agent [26] possessing anticholinesterase activity [29]. In the present paper, it was used as a source of complementary information regarding different aspects of 6ID and 6ND activity, which might not be related directly to α7 nAChR activity.

Synthetic hypaphorines inhibit heteromeric α3* nAChRs at micromolar and α9/α10 nAChRs at sub-millimolar concentrations, which might explain some of the effects outside the scope of the α7 nAChR-attributed anti-inflammatory activity (Figure 3 and Appendix A). It should be stressed, that calcium response detected in the assay described in this paper is an indirect measure of the receptor activity. Thus, the real affinity toward α7 nAChR of the synthesized compounds might deviate from the measured efficiency (potency). However, the activity of the synthesized compounds was also determined in competition with radioiodinated α-Bgt for α7 nAChR binding site (Appendix A) and it can be ranked in a good agreement with the calcium imaging results: compound 1 > 4 > 5, 9 > 6, 7, 10, 12 vs. 1 > 4, 5 > 9 > 6, 7 > 10 > 12, respectively (numbering corresponds to the Table 1).

The analysis of biological activity of 6ID, a new α7 nAChR agonist, in comparison with PNU 282987, a well-established selective agonist of this receptor subtype, was performed in vitro on human primary macrophages. It was found that PNU 282987, but not 6ID or hypaphorine, leads to a significant increase in the surface expression of CD11b and CD11c proteins (Figure 4a,b). Complement receptors CR3 (CD11b/CD18) and CR4 (CD11c/CD18) belong to the β2-integrin family and play an important role in cell adhesion and migration, as well as in phagocytosis [44].

On the other hand, PNU 282987, 6ID, and hypaphorine significantly decreased the toll-like receptor 4 (TLR4) surface expression (Figure 4c,e). TLR4 is a member of the TLR family that is recognized and activated by bacterial lipopolysaccharide (LPS), a major component of the cell wall of gram-negative bacteria. Molecular recognition of LPS by the TLR4 receptor system triggers a cascade leading to the production of pro-inflammatory cytokines initiating the inflammatory responses [45]. Reduction of cell surface TLR4 expression under the influence of PNU 282987, 6ID, and hypaphorine may alleviate inflammatory response. The most pronounced inhibition of TLR4 receptor was observed when macrophages were treated with 6ID. Similar data were published earlier, showing that treatment of HMEC-1 endothelial cells by hypaphorine from Erythrina velutina resulted in the inhibition of TLR4 expression [27]. Thus, the hyperinflammatory response can be partially blocked by these compounds due to suppressing the expression of TLR4 receptors.

Complementary to that, 6ID and hypaphorine significantly raised the surface expression of CD86, a known regulator of IL-10 anti-inflammatory response [46]. CD86 is a co-stimulatory molecule for the priming and activation of T cells. At an early stage of the immune response, CD86 is expressed in cells of primary lymphoid tissue and is constitutively expressed in the antigen-presenting cells [46]. CD86 promotes antigen presentation to T cells and regulates the anti-inflammatory response. In addition, CD86 is essential for Th2 response [47]. There are reports that acute inflammation during sepsis is associated with suppression of constitutive expression of CD86 [46,48]. In addition, some studies show that CD86 plays a role in the regulation of the inflammatory response in vivo in diseases regulated by the adaptive immune response [49]. We found that treatment of macrophages with hypaphorine and 6ID resulted in the increased expression of CD86 while PNU 282,987 had no effect, suggesting different molecular mechanisms.

Signaling pathways upon activation of macrophages by LPS were also explored in the presence of the compounds described in this communication. Previous studies have shown that PNU 282,987 [50] and nicotine [51,52] increase phosphorylation of ERK 1/2 in different types of cells. In addition, activation of the STAT3 pathway through α7 nAChR has been shown to play a critical role in the prevention of inflammation [16,53]. In our study, it is shown that the treatment of LPS-stimulated macrophages with PNU282987, hypaphorine, or 6ID leads to an increase in the phosphorylation of ERK 1/2 and STAT3 (Figure 5). Interestingly, treatment of murine macrophages with hypaphorine was reported to decrease EPK 1/2 phosphorylation [26]. These results suggest that modulation of EPK 1/2 by hypaphorines obtained from different sources may proceed differently in different types of cells, thereby regulating various kinds of biological functions.

Anti-inflammatory properties of new hypaphorine analogs were confirmed in several in vivo models. In a mice model with subplantar administration of complete Freund’s adjuvant (CFA), the compounds 6ID, 6ND, and hypaphorine demonstrated anti-oedemic properties and an analgesia in a hot plate test (Figure 6). Hypaphorine, lacking α7 nAChR-agonistic properties, had significantly lower anti-oedemic activity (Figure 6d). The latter conclusion has been confirmed in test on mice inflammation model with subplantar 1% carrageenan injections: 6ND and 6ID demonstrated significant (at *p* < 0.05 level) anti-oedemic effect in comparison to vehicle-treated animals, whereas hypaphorine was ineffective (*p* = 0.131, one-way ANOVA). Despite being ineffective against oedema, hypaphorine showed strong evidence of analgesia in algesimeter and von Frey tests (Figure 7g and Figure 7h, respectively). These results suggest that analgesic effects of hypaphorine and its analogs 6ND and 6ID depend on different molecular mechanisms, some of them not involving α7 nAChR.

Modelling of arthritis-like inflammation by introducing monoiodoacetate (MIA) through the patellar ligament into the intra-articular space of the right knee revealed anti-invalidation effects of the 6ND hypaphorine analog (improved grip strength, Figure 8d). 6ND also showed analgesic (in mechanical allodynia test, Figure 8f) and anti-oedemic (Figure 8h) effects in MIA-induced arthritis rat model.

The administration of 6ND in the model of MIA-induced arthritis at a dose of 0.26 mg/kg demonstrated a significant reduction in the morphological manifestations of synovitis and synovial hyperplasia of the right knee joint of female rats.

Thus, 6-substituted esterified hypaphorine analogs show promising results as a potential new class of anti-inflammatory agents, having an increased affinity for α7 nAChR.

## 4. Materials and Methods

### 4.1. Rational Design and Virtual Screening

A virtual library of fifty-six hypaphorine analogs was constructed using the GNU/Bash script to automatically generate simplified molecular-input line-entry system (SMILES) format [32] of all possible combinations of hypaphorine modifications from the following set: X = NO_2_, NH_2_, OH, OCH_3_, Cl, Br, I; Q = OH, OCH_3_, OC_2_H_5_, NH_2_, NHCH_3_; R = H, CH_3_, where X denotes indole ring substitution in 6th position, Q is either carboxy, amide, or ester group, and R encodes the methylation state of the ammonium nitrogen (see Figure 2b). Open Babel [33] was used to convert generated SMILES to MOL2 files and UCSF Chimera was utilized to minimize internal energy of the generated structures. Autodock Tools were used to generate PDBQT files for the designed panel of molecular structures and prepare the receptor structure (PDB 7KOX [34]) for virtual screening. In brief, water molecules except the conservative water in the orthosteric binding site have been removed, Gasteiger charges were added, and all hydrogens, with the exception of polar ones, were removed. Grid box was centered at Trp 148 located in the orthosteric binding site and playing the crucial role in the receptor activation. Grid box dimensions were set to 22 × 22 × 22 points (at spacing 0.375Å) to accommodate all possible conformations of hypaphorine analogs. Final docking results were visually inspected in UCSF Chimera (Figure 2) and the ligands which produced meaningful conformations of the complexes were selected for chemical synthesis.

### 4.2. Calcium Imaging

Mouse neuroblastoma Neuro2a cells were purchased from the Russian collection of cell cultures (Institute of Cytology, Russian Academy of Sciences, Saint Petersburg, Russia). Cells were cultured in Dulbecco’s modified Eagle’s essential medium (Paneco, Moscow, Russia) supplemented with 10% fetal bovine serum (Sigma, St. Louis, MO, USA). The cells one day before transfection were subcultured and plated at a density of 10,000 cells per well on a 96-well plate. The next day, Neuro2a cells were transiently transfected with plasmids coding α7 nAChR (human α7 nAChR-pCEP4) or its mutants, the chaperone Ric-3 (Ric3-pCMV6-XL5, OriGene, USA) and a fluorescent calcium sensor Case12 (pCase12-cyto vector, Evrogen, Russia) in molar ratio 4:1:1. Lipofectamine transfection protocol (Invitrogen, Waltham, MA, USA) was performed as recommended by the manufacturer. Transfected cells were grown at 37 °C in a CO_2_ incubator for 48 h.

Calcium immobilization assay was performed as described previously [35]. Briefly, Transfected Neuro2a cells were grown on black 96-well plates (Corning, Somerville, MA, USA) at 37 °C in a CO_2_ incubator for 72 h, then growth medium was substituted with buffer containing 140 mM NaCl, 2 mM CaCl_2_, 2.8 mM KCl, 4 mM MgCl_2_, 20 mM HEPES, 10 mM glucose; pH 7.4.

Cells were incubated with the α7 nAChR positive allosteric modulator PNU120596 (10 μM, Tocris, Bristol, UK) for 20 min at room temperature before the addition of tested compounds. Plates were measured in microplate reader Hidex Sence (Hidex, Turku, Finland) using excitation at 485 nm and emission at 535 ± 10 nm. Fluorescence peak intensity in each well was expressed as a percentage of the maximal obtained response. Data files were analysed using Hidex Sence software (Hidex, Turku, Finland). Controls were run in the presence of 4 μM α-cobratoxin.

### 4.3. Flow Cytometry

Peripheral blood mononuclear cells (PBMCs) were separated by Ficoll-Paque PLUS density gradient centrifugation. PBMCs were placed in a sterile Petri dish and incubated at 37 °C for 2 h. Unattached cells were then removed by washing with PBS and substituted with fresh complete RPMI 1640 medium. To generate monocyte-derived macrophages (MDMs), 50 ng/mL GM-CSF was added to the isolated monocytes and cultured for 6 days to differentiate them into non-polarized MDMs.

### 4.4. In Vivo Anti-Inflammatory Activity

#### 4.4.1. Animals

Specific pathogen-free outbred ICR male mice (6 to 8 weeks old, weighing 29 to 33 g) and Wistar female rats (8–9 weeks old 250 ± 25 g) were obtained from the Animal Breeding Facility of the Branch of the Shemyakin–Ovchinnikov Institute of Bioorganic Chemistry of the Russian Academy of Sciences (Pushchino). The animals were acclimatized for 2 weeks before the experimental procedures and were kept in two corridor barrier rooms under a controlled environment: a temperature of 20 to 24 °C, a relative humidity of 30% to 60%, and a 12 h light cycle. The animals were housed in Type 3 standard polycarbonate cages (820 cm2) on bedding (LIGNOCEL BK 8/15, JRS, Rosenberg, Germany), with ad libitum access to feed (SSNIFF V1534-300, Spezialdiaeten, GmbH, Soest, Germany) and filtered tap water. The mouse cages were also supplied with material for environmental enrichment, i.e., Mouse House (Techniplast, Buguggiate, Italy).

The study was conducted in AAALAC (Association for Assessment and Accreditation of Laboratory Animal Care International) accredited facility in compliance with the standards of the Guide for Care and Use of Laboratory Animals (8th edition, Institute for Laboratory Animal Research). Animal treatment procedures were approved by the Institutional Animal Care and Use Committee (IACUC) of the Branch of the Shemyakin–Ovchinnikov Institute of Bioorganic Chemistry, Russian Academy of Sciences, the experimental protocol code is no. 688/19 (date of approval: 10 January 2019).

#### 4.4.2. CFA-Induced Inflammation Test

The development of the inflammation and thermal hyperalgesia of the paw was induced by the injection of the oil/saline (1:1) CFA emulsion (Sigma-Aldrich, St. Louis, MO, lot # WH327536) into the subplantar surface of the hind paw of the ICR mice (20 µL/paw) 24 h before the measurement. Oedema was measured with an electronic calliper. Immediately after paw measurement, 6ND or 6ID were injected intramuscularly in doses: 1, 0.5, and 0.1 mg/kg. Paws were measured at 6, 12, 24, and 48 h after administration of 6ND or 6ID. After 2 h following the drug injection, the inflamed paw withdrawal or licking latencies to thermal stimulation were measured on a hot plate device (Hot Plate Analgesia Meter, Columbus Instruments) with a set temperature of 53 ± 0.1 °C and a cut-off time of 60 s.

#### 4.4.3. Carrageenan-Induced Inflammation Test

Carrageenan inflammation was induced by injecting 30 μL of a 1% carrageenan suspension (Sigma-Aldrich, lot #SLBK3896V) into the subplantar surface of the ICR mice hind paw. The oedema degree of the carrageenan-induced paw was evaluated using an electronic calliper by the volume difference of the animal’s right hind paw after (3 h from carrageenan injection) and before being hurt by carrageenan.

One hour after carrageenan injection, the drugs (hypaphorine, 6ND, or 6ID) were administered at a dose of 0.5 mg/kg. Hypaphorine and 6ID were also administered at 1 mg/kg. The control group was treated with saline (0.9% NaCl).

Antinociceptive testing was performed 2 h after the drug administration (3 h after carrageenan injection).

Nociceptive testing was performed by utilizing the von Frey test and Paw pressure test.

#### 4.4.4. Von Frey Test

The animals were placed in 20 cm × 20 cm Plexiglas boxes equipped with a metallic mesh floor. Animals were allowed to habituate themselves to their environment for 15 min before the test. The electronic von Frey instrument (model BIO-EVF4; Bioseb, Vitrolles, France) was used to vertically stimulate the center of the rat hind paw with increasing intensity until the hind paw was lifted, the withdrawal threshold was automatically displayed on the screen. The paw sensitivity threshold was defined as the minimum force required to elicit a robust and immediate withdrawal reflex of the paw. Spontaneous movements associated with locomotion were not considered as a withdrawal response. Measurements were repeated 3 times and the final value was obtained by averaging the 3 measurements.

#### 4.4.5. Paw Pressure Test

Mechanical hyperalgesia was measured as a paw withdrawal response to a gradual increase of mechanical pressure applied by the Rodent pinchers—analgesia meter (model BIO-RP-M; BioSeb, Vitrolles, France). The influence of stimulation on each hind paw was recorded three times. The maximum force applied to the paw was recorded as the grams (g) of force on the dynamometer.

#### 4.4.6. Monoiodoacetate-Induced Arthritis Model

Before the study, 26 female Wistar rats (250 ± 25 g), 8–9 weeks old, were divided into cages of four so that the average body weight did not differ between groups. Four groups of animals were formed: control and three groups with modelling of inflammation by introducing monoiodoacetate (MIA). Three groups of animals were injected with 3 mg MIA in 50 μL of 0.9% sodium chloride through the patellar ligament into the intra-articular space of the right knee using a 29-G needle. Control rats received an equivalent volume of 0.9% sodium chloride.

Assessments of inflammation in vivo and functional tests were conducted on days 8 and 16 after arthritis model induction. Thermal and mechanical hypersensitivity and pain-induced articular disability were tested in this model. Body weight gain and paw oedema were also monitored during the experiment.

The inflamed paw withdrawal or licking latencies to thermal stimulation were measured on a hot plate device (Hot Plate Analgesia Meter, Columbus Instruments) with a set temperature of 53 ± 0.1 °C and a cut-off time of 60 s.

Mechanical hypersensitivity was measured by the electronic von Frey instrument (model BIO-EVF4; Bioseb, Vitrolles, France) used to vertically stimulate the center of the rat hind paw with increasing intensity. Measurements were repeated 3 times and the final value was obtained by averaging the 3 measurements.

Behavioral assessment of movement-caused pain was carried out in a hind limb grip strength test using a Grip Strength Meter (Columbus Instruments, Columbus, OH, USA) consisting of a wire mesh frame connected to the gauge. Rats were restrained and allowed to grasp the wire mesh frame with their hind paws and then were pulled backwards until the grip released. Three measurements were conducted with an interval of 30 s for the averaged grip strength calculation.

### 4.5. Statistical Analysis

The R statistical programming environment “https://www.r-project.org/” (accessed on 16 June 2023) was used to analyze the results. Base R graphics and ggplot2 [54] graphical library were used for the data visualisation. Dose-response curves were fitted using “nlm” function and the following equation: response ~ (100/1 + (EC_50_/x)^n), where x denotes base 10 logarithm of ligand concentration in moles per liter (normalized to 1M) and n is the Hill coefficient. Libraries “rgl” “https://CRAN.R-project.org/package=rgl” (accessed on 16 May 2023) and “barplot3d” “https://cran.r-project.org/src/contrib/Archive/barplot3d/” (accessed on 16 May 2023) were used to generate the 3D bar chart shown on Figure 2b.

One- and two-way ANOVA with Tukey post hoc test were used to examine the significance of the differences between groups. Repeated measures ANOVA was utilized to explain longitudinal data. In all cases *p* < 0.05 was interpreted as significant.

### 4.6. Two-Electrode Voltage-Clamp

Recordings were performed using Axopatch 200 amplifier in two-electrode voltage-clamp setup (Molecular Devices, LLC, San Jose, CA, USA) on oocytes removed from mature *Xenopus* frogs. Two to three days before taking recordings, oocytes were injected with in vitro transcribed RNA, coding α9 and α10 nAChR and kept at 18 °C in ND96 electrophysiology buffer solution (5 mM HEPES/NaOH at pH 7.6 and 18 °C, 96 mM NaCl, 2 mM KCl, 1.8 mM CaCl_2_, 2 mM MgCl_2_).

### 4.7. Radioligand Competition

Tested compounds were incubated with GH4C1 cells expressing α7 nAChR in 50 μL of 20 mM Tris-HCl buffer pH 8.0, containing 1 mg/mL BSA (binding buffer) for 30 min. Then, 0.5–0.9 nM of mono-iodinated 125I-αBgt was added for 5 min and cells suspensions were applied to GF/C glass filters (Whatman, Maidstone, UK) presoaked in 0.1 mg/mL BSA, and unbound radioactivity was removed from the filter by washing (3  ×  3 mL) with 20 mM Tris-HCl buffer, pH 8.0 containing 0.1 mg/mL BSA (washing buffer). Nonspecific binding was determined by preliminary incubation of GH4C1 cells with 30 µM α-cobratoxin. The bound radioactivity was determined using a Wizard 1470 Automatic Gamma Counter.

## 5. Conclusions

In this paper, we describe a panel of synthetic analogs of natural marine product 6-bromohypaphorine with elevated potency toward α7 nAChR (as can be deduced from intracellular calcium rise and competition with radioactive α-bungarotoxin), demonstrating the pronounced anti-inflammatory and analgesic activities. Introduction of iodine instead of bromine, esterification of carboxy group, and changing chirality from L isomers to D was found to be the most productive strategy.

## Data Availability

The data presented in this study are available on request from the corresponding author.

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
