# Peer review of "Analogs of 6-Bromohypaphorine with Increased Agonist Potency for α7 Nicotinic Receptor as Anti-Inflammatory Analgesic Agents"

_marinedrugs, 2023, doi:10.3390/md21060368_

Round 1

Reviewer 1 Report

Using molecular modelling of the α7 nicotinic acetylcholine receptor (nAChR), the investigators developed analogues of L-6-bromohypaphorine with increased potency. Fourteen designed analogs were synthesized and tested in vitro by calcium fluorescence assay on the α7 nAChR expressed in neuro 2a cells, methoxy ester of D-6-iodohypaphorine (6ID) showing the highest potency (EC50 610 nM), being almost inactive toward α9α10 nAChR. Anti-inflammatory activity was established in macrophages. 6ID administration in doses 0.1 and 0.5 mg/kg decreased carrageenan-induced allodynia and hyperalgesia in rodents, reflecting its anti-inflammatory action. The methoxy ester of D-6-nitrohypaphorine demonstrated anti-oedemic and analgesic effects in arthritis rat model at i.p doses 0.05-0.26 mg/kg. Tested compounds showed excellent tolerability with no acute in vivo toxicity in dosages up to 100 mg/kg.

Overall, this paper represents an outstanding body of work that is entirely suitable for publication in Marine Drugs. In particular, the identification of an analogue (6ID) that is more than 100-fold more potent than the parent compound (L-6-bromohypaphorine) is a major development.

There is one area where improvement is sorely needed and that is in the figures that appear to have been hastily assembled. Many of the figures have panels that differ in size and furthermore, the X- and Y-axes are not aligned. These include figures 4, 5, 6, 7 and 8. In multiple figures the addition the labels of the panels (a, b, c, etc.) appear to align with the panels below the ones that they actually refer to. This problem is seen in figures 1, 2, 4, 5, 6, 7, 8, and 9. These problems need to be fixed because they currently look scrappy and detract from an otherwise excellent paper.

The data in Figure 4, may be statistically significant, but the biological significance of such small changes eludes this reviewer. Does this figure really add anything to the paper? This criticism could also be focused on phospho STAT-5 in figure 5.

Minor point

Keywords should include: Hypaphorines and α7 nicotinic acetylcholine receptor (nAChR). This reviewer would consider these more important than: “rational drug-design”

Author Response

There is one area where improvement is sorely needed and that is in the figures that appear to have been hastily assembled. Many of the figures have panels that differ in size and furthermore, the X- and Y-axes are not aligned. These include figures 4, 5, 6, 7 and 8. In multiple figures the addition the labels of the panels (a, b, c, etc.) appear to align with the panels below the ones that they actually refer to. This problem is seen in figures 1, 2, 4, 5, 6, 7, 8, and 9. These problems need to be fixed because they currently look scrappy and detract from an otherwise excellent paper.

We thank the Reviewer for this comment. Misalignment happened due to partial incompatibility of word processing software with the journal template. To avoid misalignment, we re-compiled figures to avoid possible shifts of labels and panels.

The data in Figure 4, may be statistically significant, but the biological significance of such small changes eludes this reviewer. Does this figure really add anything to the paper? This criticism could also be focused on phospho STAT-5 in figure 5.

To elucidate the possible mechanisms of the anti-inflammatory effect of the studied compounds, we evaluated the expression both of various macrophage membrane proteins and intracellular signaling proteins. The changes in the expression of some proteins were not so pronounced. However, it should be noted that similar effects have been previously described for cholinergic anti-inflammatory agents such as nicotine and GTS-21 (doi:10.1016/j.bcp.2009.06.096, DOI: 10.1007/s11481-017-9760-7, DOI: 10.1007/ s11481-017-9760-7, 
doi.org/10.1093/infdis/jit669). Apparently, the complex effect on the expression of several macrophage proteins involved in inflammatory processes, as well as that of the signaling proteins, is sufficient for the manifestation of the anti-inflammatory activity of our synthetic compounds.

Minor point

Keywords should include: Hypaphorines and
α7 nicotinic acetylcholine receptor (nAChR). This reviewer would consider these more important than: “rational drug-design”

These keyword were included and in general this list of keywords has been amended.

Reviewer 2 Report

The current study synthesized 6 new analogous of hypaphorine and performed in-vitro and in-vivo assays to understand how these analogues elicit their effect via a7 nAChR. While the authors use wide-ranging techniques, there are too many low loose ends. Below are some of the comments. The manuscript needs significant re-writing with emphasis on a clear reasoning for connecting experiments.

Title is misleading – Affinity doesn’t correlate EC50 directly. I would rather use potency any way. Even so the functional assay used in the study doesn’t not reflect true bind-gate of the nAChRs.

Figure 2: One fundamental shortcoming of using microplate reader to measure intracellular Ca2+ as a functional assay for ion-channel activity is that it does not reflect ions conducted through the channel, because of fast conduction. Most of the observations at this stage is from secondary signaling triggered from release of intracellular Ca2+ stores. While it is a functional assay, not sure if comparing EC50’s is a true reflection of nAChR activity, with the steep hill slopes sometimes with no data points in them.

Figure 3: Were the compounds applied without the presence of nicotine on a3 nAChRs. Even partial agonists can appear to be inhibitor when co-applied with a strong agonist, such as nicotine, in this case. I would suspect you would see Ca2+ influx even when applying the ligands alone, because its secondary signaling anyway. Intracellular signaling not well corelated with full or partial agonists.

Figure 5: Erk1/2 phosphorylation in most cell-types is one of the first markers to change that happens in the matter of minutes. Here, the time scale is in the order of hours. Can authors explain the justification of the time scale used in the study. 

The manuscript needs significant attention on the construction of sentences 

Author Response

1) Title is misleading – Affinity doesn’t correlate EC50 directly. I would rather use potency any way. Even so the functional assay used in the study doesn’t not reflect true bind-gate of the nAChRs.

We thank the reviewer for the clarification and changed the word “affinity” in the title for “potency’. Indeed, dose-response relationships under the described conditions don't necessarily reflect the receptor activity but rather show more complex response of the receptor-expressing cell to the application of the tested compound. However, in our experience, the system used here (which includes neuro 2a cells with the expressed alpha7 nAChR) allows us to rank tested compounds in the efficiency of the increasing the intracellular calcium measured by the fluorescence method. To confirm that the observed effect is indeed due to their binding to the alpha7 nAChR, the radioligand binding competition experiment was performed on synthetic hypaphorines at 10 microM (Figure S2). All tested compounds have shown the ability to displace 125I-alphaBgt from the orthosteric binding site of alpha7 nAChR. The activity rank corresponds to the ranking obtained in calcium imaging experiments: 1 > 4 > 5, 9 > 6, 7, 10, 12 vs. 1 > 4, 5 > 9 > 6, 7 > 10 > 12, respectively (numbering corresponds to the Table 1). In our previous studies (Shelukhina et al., 2017, reference 35 in the manuscript), it was confirmed that EC50 values calculated from the dose-response relationships in this assay are in fact reasonably informative and sensitive to the minor changes in the agonist affinity.

2) Figure 2: One fundamental shortcoming of using microplate reader to measure intracellular Ca2+ as a functional assay for ion-channel activity is that it does not reflect ions conducted through the channel, because of fast conduction. Most of the observations at this stage is from secondary signaling triggered from release of intracellular Ca2+ stores. While it is a functional assay, not sure if comparing EC50’s is a true reflection of nAChR activity, with the steep hill slopes sometimes with no data points in them.

We tried to address this concern in response to the previous point. The reason behind using the calcium imaging for compounds screening is not only the convenience and throughput, but also the significance of calcium for the physiological activity manifestations. Since alpha7 nAChR predominantly conducts calcium, the hit compounds should be considered with respect to their ability to evoke calcium response instead of traditional observations of current amplitudes in electrophysiology.
Nevertheless, in our experience, the assay used here gives reasonable correlation with the affinity of agonists [35] as already mentioned to the criticism in the previous comment.

3) Figure 3: Were the compounds applied without the presence of nicotine on a3 nAChRs. Even partial agonists can appear to be inhibitor when co-applied with a strong agonist, such as nicotine, in this case. I would suspect you would see Ca2+ influx even when applying the ligands alone, because its secondary signaling anyway. Intracellular signaling not well corelated with full or partial agonists.

We are grateful to the Reviewer for the valuable insights into possible mechanisms underlying the observable properties. We did not detect significant calcium rises in response to 6ID and 6CF applications (Figure 3 now is amended accordingly). We understand that in the case of weak partial agonism, the low amplitude responses can be missed due to sensitivity limitations and such partial agonists would antagonize nicotine (full alpha3 nAChR agonist). These considerations have been incorporated into the respective subsection.

We also deleted the results on 6ND and 6NAM that were not tested "solo" on alpha3 nAChRs yet.

4) Figure 5: Erk1/2 phosphorylation in most cell-types is one of the first markers to change that happens in the matter of minutes. Here, the time scale is in the order of hours. Can authors explain the justification of the time scale used in the study.

Indeed, activation of monocytes/machrophages by LPS leads to rapid activation of ERK1/2. On the other hand, high levels of ERK1/2 phosphorylation are maintained over a long period under LPS exposure, peaking after 1 hour of exposure (DOI: 10.3109/10715762.2011.564165, DOI: 10.1128/AAC.48.6.1974-1982.2004, DOI: 10.4049/jimmunol.1001739, 
doi.org/10.3389/fmicb.2017.01393doi.org/10.1186/1471-2172-8-24). Based on the results of previous studies, a time period in the order of hours was chosen in this work.